# Comparing sentencing judgments of judges and laypeople: The role of justifications

**Eiichiro Watamura** [ID]*, **Tomohiro Ioku**

Graduate School of Human Sciences, Osaka University, Suita, Osaka, Japan

* watamura@hus.osaka-u.ac.jp

## Abstract

There is a lack of understanding concerning the differences between laypeople's and professional judges' conceptions of justifications for sentencing. We conducted an online quasi-experimental study with 50 active judges and 200 laypeople. Participants were presented with a vignette describing severe child abuse leading to fatality and were asked to indicate a term of imprisonment for the father and the justification they would consider relevant when deciding on the sentence. A two-factor analysis of variance showed that laypeople disproportionately favored retribution compared to judges. This was reflected in the judges' higher scores for the other three justifications (incapacitation, general deterrence, rehabilitation). The Likert scales failed to detect any such differences. Furthermore, imprisonment terms given by judges were shorter than those given by laypeople. These results support the hypotheses that judges balance multiple justifications and find a shorter sentence that is appropriate; their lesser bias toward retribution supports the notion that judges should be balanced and fair-minded.

## Introduction

This study examines "retributivism" versus "utilitarianism"—a jurisprudential issue regarding the justification of punishment—from the perspective of the differences between experts and non-experts. Retributivism implies that the essence of punishment is to provide just deserts for crime; it seeks to offset crime through punishment [1–3]. Utilitarianism is based on the idea that punishment should serve the function of preventing future crimes [1,4]. It, therefore, seeks to deter recidivism and potential offenders from committing crimes. In the past two decades, experiments and surveys with mock juries have led to a dramatic increase in social psychological research on laypeople's sentencing decisions [5,6]. These studies have suggested that laypeople prefer retributive focused philosophies, as opposed to the human image of living in a civilized society that is receptive to the mistakes of the accused. Laypeople tend to resolve moral imbalances by harsh punishment commensurate with the severity of the crime, without much consideration of the practical effects of the punishment [1–4,7–13]. Prior studies have found positive arguments supporting the fact that retribution serves as the perpetrator's compensation to society [14] and that it should not be neglected to maintain public confidence in the judiciary [15]. Many future-oriented arguments claim that retribution should also aim for

**Data Availability Statement:** All data from this study can be found at the following link: https://osf.io/sq9gf/.

**Funding:** Initial of the author who received each award: E.W Grant number:19K14358 The full name of each funder: JSPS KAKENHI URL of each funder

website: https://www.jsps.go.jp/english/e-grants/index.html The funders had no role in study design, data collection and analysis, decision to publish, or preparation of the manuscript.

**Competing interests:** The authors have declared that no competing interests exist.

restorative justice and social inclusion of the accused [16–18]. Tsoudis [19] found extreme differences concerning the perceptions of criminal justice issues between college students majoring in criminal justice and those majoring in other subjects. This finding suggests that laypeople's retributive one-sided judgement may result from an inadequate understanding of crime and criminal justice. It may also be due to beliefs and ideologies influenced by the media [20,21] owing to this lack of understanding. However, it is unclear whether laypeople are harsher on the accused because of their ignorance. A comparison with judges is needed to understand laypeople's view of punishment and to develop practical and theoretical research on legal systems and education in the future.

## Justification

The critical factor that systematically influences sentencing decisions is the justification or rationale that determines the severity of the punishment [8]. Justification, as a basic decision-making strategy, influences the selection of information and the evaluation of its importance [2,3]. For example, the defendant's low rehabilitation potential (e.g., criminal record) is insignificant in a judgment based on retribution; however, a judgment based on rehabilitation suggests the need for a strong educational effect, and so, relatively heavy penalties are imposed. Conversely, the victim's pain and suffering may not be direct factors in influencing a rehabilitation-based decision, but they may be necessary elements in a retribution-based decision to mediate the assessment of malice and increase the severity of the punishment. In this study, to highlight the judgments of judges as professionals, we tested whether there is a difference between justification and the length of imprisonment deemed "appropriate" by having judges and laypeople make sentencing decisions for the same child abuse death case. Child abuse creates an extremely malicious impression because the accused not only fails to fulfill their parental duties but also harms the overwhelmingly vulnerable victim [21]. Such viciousness is emphasized by the media, which assemble and report abuse cases as tragic episodes rather than social problems [22–24]; in cases where the victim has died because of abuse, laypeople are likely to be strongly aroused because of the severity of both the consequences and the perpetrator's malice [25]. Especially noteworthy for our study, while child death evokes an extremely strong desire for retribution, the utilitarian justification for rehabilitation can equally well be emphasized—the infliction of punishment may make such "unfit parents" remorseful, preventing re-offense. Incapacitation, another utilitarian justification that validates long-term isolation in prison until the risk of recidivism is reduced, could also be considered. Justification, which sends a demonstrative message to society, decreasing the risk of spreading abuse cases, may also be considered. Hence, child abuse deaths strongly evoke strict retribution, but with room for various justifications. This makes it an ideal touchstone for this study, which seeks to distinguish between judges and laypeople. Unlike laypeople, professional judges are accustomed to balancing retribution and utilitarian justification because of their professional experience [26,27]; they are trained to be less influenced by bias toward retribution, despite the cruel outcome of a child's death, balancing various justifications instead. Moreover, because of this balanced judgment, their sentences are likely shorter than those of laypeople. Thus, this study aimed to clarify the differences between judges and laypeople through a quasi-experiment—no random assignment intervention experiment—comparing sentencing for child abuse deaths and justifications for the sentences chosen. We tested the following hypotheses.

In judgments of malicious child abuse and deaths:

*Hypothesis 1*: Judges are not biased toward retribution but give importance to other justifications as well when sentencing.

*Hypothesis 2*: Judges' sentences are shorter than those of laypeople.

The severity and urgency of child abuse have been recognized worldwide [28,29]. These have been accelerated by COVID-19, making the home environment more restrictive and imposing greater restraints on patrols [30–33]. By clarifying the perceptions of both professionals and amateurs in the criminal justice system, our study provides significant practical insights for decision-making in trial and criminal justice policy concerning child abuse cases.

## Materials and methods

Judges and laypeople were recruited from the panel of 2.2 million people representing the Japanese adult population (20 years and older) registered with the Internet survey company Rakuten Insight, Inc. After receiving an e-mail invitation, they read the instructions online, provided informed consent by clicking the "Agree and Start Questionnaire" button (i.e., electronic written consent), and participated in the online experiment on a transitional screen. The sample of judges consisted of 50 judges (45 males, $M_{age}$ = 51.74, $SD$ = 9.08) who indicated that they were currently serving in Japanese courts from a pool of professionals such as lawyers and doctors. Because there were too few judges, we included judges with any type of background demographics to ensure we could recruit 50. We randomly selected 200 laypeople aged between 20 and 69 years from the same firm's panel so that there were equal numbers of males and females (100 males, $M_{age}$ = 44.57, $SD$ = 13.87). Gratuities were the equivalent of US$0.2 in shopping points that could be used in the company's affiliated groups and were paid after their participation.

This study was approved in writing by the Ethics Review Committee of the Department of Behavioral Sciences, Graduate School of Human Sciences, Osaka University (Approved number: HB020-023). At the beginning of the study, participants were presented with a fictional vignette of approximately 450 words describing the death of a six-year-old boy subjected to severe violence by his father for 10 months (Appendix). Participants could read the vignette at any time until they had completed all the responses asked in the questionnaire. They were asked to answer the following question in a free-text format before responding to the other questions:

If you were to ask questions of or express opinions to the accused (the father of the child victim), what would you say?

We did not analyze the responses to this question, as it was designed to allow participants to ponder the content of the vignette; however, it confirmed that nearly every participant responded to the question (raw data uploaded to Open Science Framework).

### Justification

Next, participants were asked to enter a number from 0 to 100 adjacent to each justification and decide the percentage they considered it necessary in determining the defendant's punishment. Numbers had to be assigned to all justifications such that they totaled 100 (an error message was displayed if the percentage was not exactly 100). The following four types and descriptions of the justifications were taken from a study by Berryessa: retribution, incapacitation, general deterrence, and rehabilitation [34]. The order of presentation was randomized for each participant.

**Retribution.**   Retribution relies on the idea that for justice to be served, an offender deserves to be punished in a manner proportionate to the severity and moral heinousness of the committed crime.

**Incapacitation.** Incapacitation aims to remove offenders from society to protect the public from future unlawful behavior.

**General deterrence.** Deterrence attempts to prevent the future committal of crimes through the threat of future punishments that outweigh an individual's motivation to commit future criminal acts.

**Rehabilitation.** Rehabilitation seeks ways to actively reform and address the underlying reasons for an offender's criminal behavior to ensure that an individual will not re-offend.

Most previous studies used a Likert scale to evaluate these justifications. However, while punishment justifications are philosophically at odds and often incompatible in practice, the Likert scale allows participants to rate high agreement with all of them (i.e., acquiescence-response; [35]). This strong endorsement of all justifications creates difficulties in identifying the relative importance of each [36]. Based on our theoretical definition that the psychological construct of sentencing is a hybrid of multiple justifications [37,38], we deemed it more appropriate to examine the relative weighting of justifications on a ratio-type scale. We also asked participants to rate their justifications on Orth's [39] Likert-type scale, which included offender deterrence (discouraging their intention to re-offend) and positive general prevention (deterrence based on the indirect effects of punishment, such as confirming social values and restoring trust in the judiciary), in addition to the four justifications listed above. Participants were asked to rate the randomly presented items (e.g., "to even out the wrong that the offender had done") on a scale from 0 (not important at all) to 5 (very important). There were two indices for each justification (12 total), that were averaged to obtain six indices.

**Sentencing decisions.** After rating the justifications on two scales, the participants were asked about sentencing and to enter the number of years that they believed the perpetrator should spend in prison. In a mock jury experiment with the sentence as the dependent variable, participants' responses can be too extreme without a reference point [40]. To prevent extreme responses, we shared a requested sentence of 13 years.

*The prosecutor asked for a prison sentence of 13 years for the perpetrator. Do you think the sentence should be longer? Do you think it should be shorter? Write down the number of years that you intuitively think are appropriate; please answer in the range of 3 to 20 years* [the legal range of prison terms for the offense].

## Results

All analyses were performed using the statistical software HAD [41].

### Justification

The results of the ratio-type scale showed that a significantly large proportion of laypeople (48.0%) chose retribution over the other three justifications. The judges also had the highest proportion of responses for retribution (35.0%), but the difference between the four justifications was controlled (Fig 1). A two-factor analysis of variance (ANOVA) was conducted with the groups (judge vs. laypeople) as the between-participants factor and justification (four types) as the within-participants factor. The results showed that the interaction between the factors was significant ($F(3, 744) = 5.20$, $p = .004$, partial $\eta^2 = .21$), reflecting group differences in the proportion of importance placed on justification. To correct the family-wise error rate, multiple comparisons were conducted using the Bonferroni correction. The $p$-values were corrected in order of absolute t-values. The results showed that retribution accounted for a larger percentage of overall justifications among the laypeople ($p$s $< .001$), whereas the other three

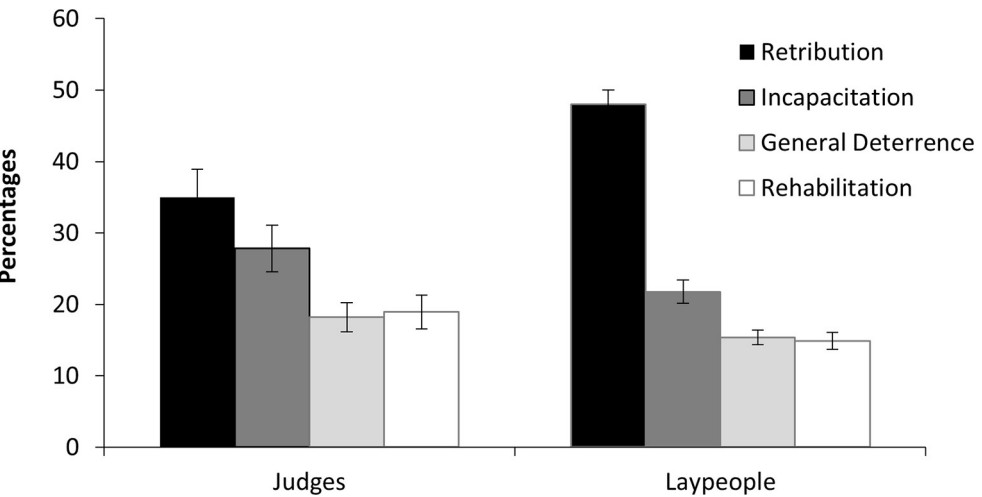

**Fig 1. Importance scores and standard error for justification on a ratio-type scale.**

factors accounted for a larger percentage among the judges ($p < .001$: incapacitation, $p = .09$: general deterrence, $p = .01$: rehabilitation). These results support Hypothesis 1, as they indicate that the judges chose other justifications more than laypersons did in deciding the sentence. The main effect of justification was also significant ($F(3, 744) = 36.26$, $p < .001$, partial $\eta^2 = .13$), and was higher in the order of retribution > incapacitation > rehabilitation ≈ general deterrence. There were significant differences in all combinations except between rehabilitation and general deterrence ($ps < .004$).

As a reference, we examined the rating of justifications as measured by the Likert scale (see Fig 2). The results showed that all the justifications were above the theoretical midpoint (3.5).

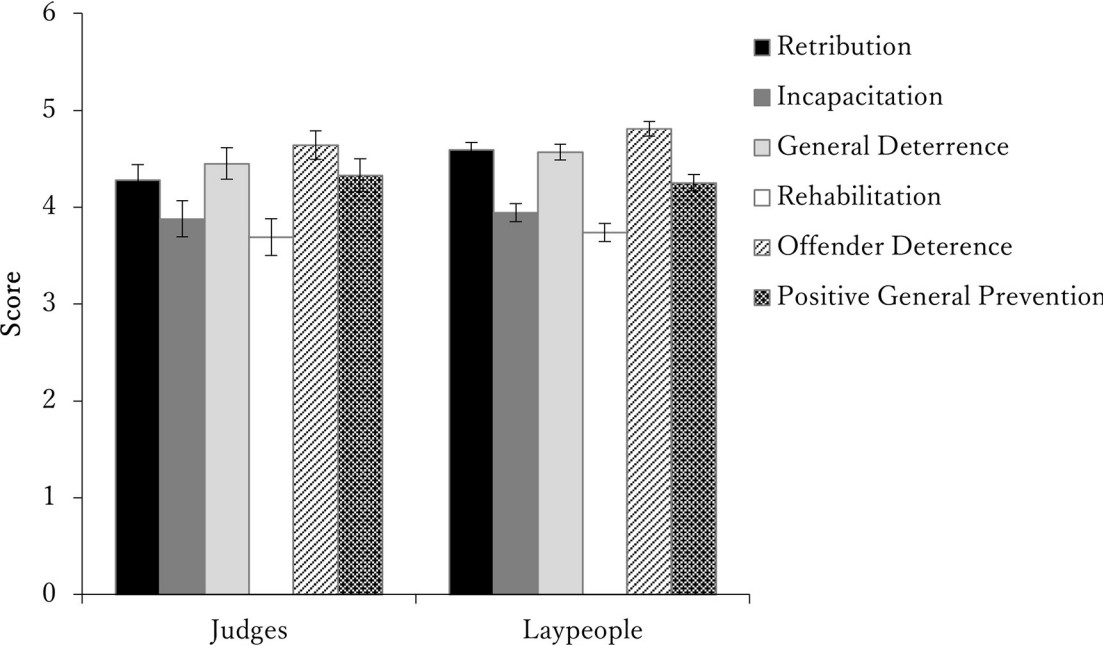

**Fig 2. Importance score and standard error for justification on the Likert scale.**

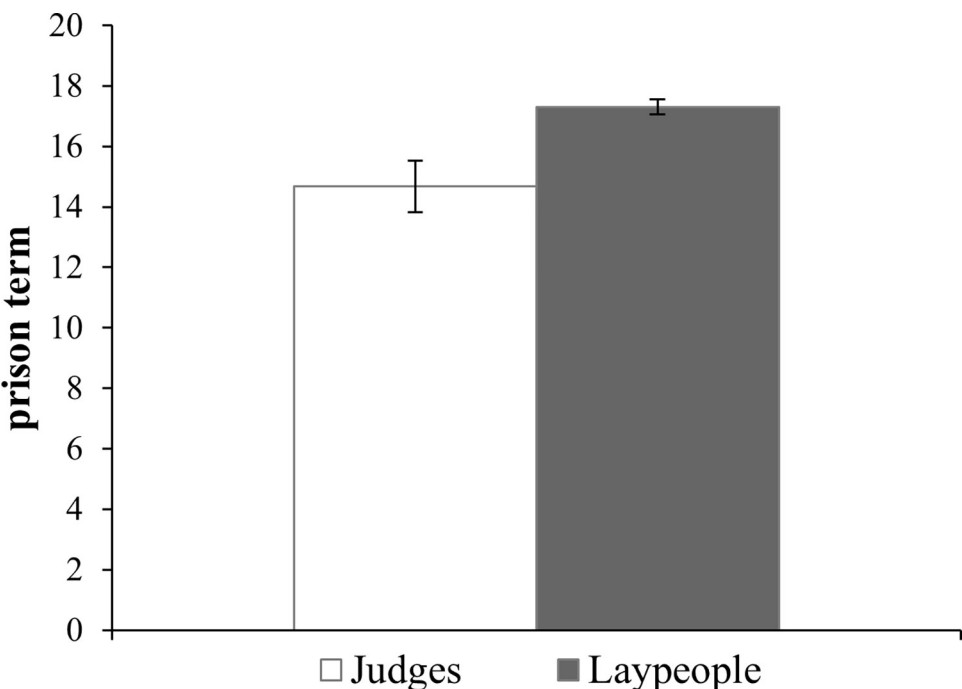

**Fig 3. Years of imprisonment suggested (error bars represent the standard error).**

A two-factor ANOVA with the two groups (judge vs. laypeople) as a between-participants factor and justification (a total of six, including offender deterrence and positive general prevention, in addition to the four justifications in the ratio-type scale) as the within-participants factor confirmed that there was no interaction ($F(5, 1240) = 0.79$, $p = .539$, partial $\eta^2 = .003$) and only a simple main effect of justification ($F(5, 1240) = 27.25$, $p < .001$, partial $\eta^2 = .99$). The effects of the justifications ranked as follows (in terms of their mean scores): offender deterrence (4.73), general deterrence (4.51), retribution (4.44), positive general prevention (4.29), incapacitation (3.91), and rehabilitation (3.72). Most scores were highly clustered, with significant differences only between the first and second and the fourth and fifth positions.

### Sentencing decisions

For both groups, the most frequent sentence was 20 years, the upper limit of the response (57.0% for laypeople and 48.0% for judges). Thus, although most of them judged that the defendant should be given a heavier sentence than the 13 years sought; on average, judges' sentences were shorter than those of laypeople ($M_{judge} = 14.68$, $SD = 6.04$ vs. $M_{laypeople} = 17.3$, $SD = 3.54$) (see Fig 3). An unpaired t-test (Welch test) confirmed that this difference was significant ($t(248) = -3.99$, $p < .001$, $d = -0.63$), supporting Hypothesis 2, which states that judges' sentencing decisions are shorter than those of laypeople.

### Discussion

In this study, we tested whether there are differences between judges and laypeople regarding sentencing of a parent accused of child abuse fatality and the justification for that sentencing. The results of the ratio-type scale of justification showed that, in contrast to laypeople heavily biased toward retribution, judges were less biased and emphasized other justifications,

supporting Hypothesis 1. Even in the case of a cruel child abuse fatality, judges did not fail to consider utilitarian justifications in addition to retribution. In particular, their intention to declare the accused an "unfit parent" to keep them from re-offending, through education or by isolating them from society, was clearly expressed. These judgments of the judges showed remarkable professionalism, even when the responses were anonymously reported. Although the analysis of published sentencing remarks showed that judges are less supportive of retribution than laypeople [12], our study found that their lack of bias toward retribution reflected their true intentions and not social pretense in the face of pressure—the notion that a judge should be balanced and think fairly. The observed results suggest that there were also differences in the sentences due to considerations of the justification. The judge's balanced decision-making strategy, not biased toward retribution, meant the defendant received a shorter sentence, averaging 14.68 years. These results support Hypothesis 2 but need to be interpreted carefully. The maximum frequency was similar for judges and laypeople at 20 years, but the variance was considerably higher for judges (6.04 vs. 3.54). The judges who knew the distribution of sentences in similar cases had less disparity (see [42–44]). The difference in prison terms shown in this study can be explained by some judges deeming a much shorter sentence appropriate. Although the judges were grouped into one category for this study, types of judges must be considered for future studies, that is, those who are relatively close to laypeople and those who are not. They can be placed on a continuum, ranging from those more similar to laypersons in their decision-making on one end and those most dissimilar at the other. The latter may reflect the character of a judge as a professional.

Furthermore, it is important to place this study in a practical context. Although child abuse is an urgent issue requiring careful consideration, public confidence in the judiciary may diminish if legal experts take the lead in judging defendants leniently at trials or making criminal policy decisions, as there are indications that their perceptions regarding the punishment of defendants differ from those of non-experts [15]. It would be thus necessary for judges to actively explain alternative viewpoints on child abuse cases, based on their professional experience, to laypeople, who are more likely to emphasize the importance of retribution and the necessity of severe punishment. That would help in building consensus with laypeople and retaining their confidence in the judiciary.

The ratio-type scale employed in this study revealed that judges and laypeople rated justifications differently. However, the scores of the Likert scale were clustered at the high end of the range, which was at odds with the theoretical prediction that "the public attaches the highest importance to retribution," which has been robustly demonstrated by previous studies (e.g., [1]). Despite this incident of child abuse fatality, the finding that retribution was ranked third out of six justifications suggests that the Likert scale may not have been an accurate measure of reality. The Likert scale measured the attitude that included a rational calculation that "utilitarian justifications must also be considered." Thus, the problem of distinguishing which justification is emphasized when awarding punishment is likely to continue to smolder as a methodological limitation of the Likert scale, especially where justifications are compatible with each other. In this study, based on the theoretical basis of the hybridity of punishment justification [37,38], we developed an incompatible ratio-type scale in which rating one judgment higher reduces the ratings for other judgments and confirms its usefulness. Due to the difficulty of introspecting the justification of punishment [43], it may be difficult to respond to a ratio-type scale with more than four or six justifications, but an examination using a minimally constructed ratio-type scale may provide a reliable litmus test to examine the important, yet unanswered question, of how judges and laypeople balance retribution with other justifications in different cases.

## Limitations

There are some limitations to this study. As in other social psychological studies, we asked the participants to make judgments with a vignette as the experimental subject. Although there is evidence that vignette studies predict courtroom performance [44], they are still inferior in quality and quantity to the materials seen in actual courtrooms. Future studies will thus need to examine judgments based on materials closer to reality. The procedure of considering questions for or opinions of the accused was a device designed to make participants read the vignettes carefully, which encouraged a conscious-controlled process [45] and may have made participants' judgments more utilitarian. If this procedure had not been included or had been cognitively loaded, their judgments (especially those of the laypeople) might have been more biased toward retribution. Although this study clarified the differences between judges and laypeople, it only showed the possibility that the differences are based on professional knowledge and experience; further, they did not examine the specifics of what knowledge is effective. Whether such knowledge and experience can lead to balanced judgments in terms of criminal justice also remains unclear. Thus, this question needs to be examined in future studies. It would also be necessary to consider justifications other than retribution, incapacitation, general deterrence, and rehabilitation. Although it was reasonable to compare judges and laypeople by focusing on these four justifications, used for comparison in most of the previous studies, the differences between judges and laypeople may also be seen in other justifications [12]. For instance, the restoration justification is legal justice, considerably different from justice achieved through punishment [46], aiming at a sophisticated justice that is typical of civilized society, distinct from retributive focused philosophies.

## Conclusion

This study aimed to understand the differences between laypeople and professional judges in terms of justification for sentencing using a vignette of a parent accused of child abuse fatality. The results indicate that both hypotheses of the study were supported; judges were unbiased toward retribution but considered other justifications important when passing the sentences, which were shorter than those of laypeople. Thus, judges and laypeople differ in their emphasis on justification and sentencing, supporting the possibility that this is due to differences in the understanding of crime and criminal justice. However, future research will need to examine the core question of why this understanding weakens the bias in favor of retribution and strengthens the support for utilitarian justification.

## Supporting information

**S1 File.**
(DOCX)

## Acknowledgments

We are grateful to Mr. Ryu Ishii, chief public prosecutor, for his helpful insights regarding this manuscript.

## Author Contributions

**Conceptualization:** Eiichiro Watamura.

**Data curation:** Eiichiro Watamura, Tomohiro Ioku.

**Formal analysis:** Tomohiro Ioku.

**Funding acquisition:** Eiichiro Watamura.

**Investigation:** Eiichiro Watamura.

**Project administration:** Eiichiro Watamura.

**Supervision:** Eiichiro Watamura.

**Writing – original draft:** Eiichiro Watamura.

**Writing – review & editing:** Tomohiro Ioku.

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
