## [Decision Letter · Decision Letter 0]

6 Sep 2022

PONE-D-22-13456Comparing sentencing judgments of judges and laypeople: the role of justificationsPLOS ONE

Dear Dr. Watamura,

Thank you for submitting your manuscript to PLOS ONE. After careful consideration, we feel that it has merit but does not fully meet PLOS ONE’s publication criteria as it currently stands. Therefore, we invite you to submit a revised version of the manuscript that addresses the points raised during the review process.

 Please note that we have only been able to secure a single reviewer to assess your manuscript. We are issuing a decision on your manuscript at this point to prevent further delays in the evaluation of your manuscript. Please be aware that the editor who handles your revised manuscript might find it necessary to invite additional reviewers to assess this work once the revised manuscript is submitted. However, we will aim to proceed on the basis of this single review if possible. The reviewer has identified some important opportunities to improve the manuscript, including clarification of aspects of the statistical analyses and provision of additional context for your study. Please respond carefully to all of the points the reviewer has raised when preparing your revisions.

We look forward to receiving your revised manuscript.

Kind regards,

Jamie Males

Editorial Office

PLOS ONE

Journal Requirements:

Reviewers' comments:

Reviewer's Responses to Questions

**Comments to the Author**

1. Is the manuscript technically sound, and do the data support the conclusions?

Reviewer #1: Yes

2. Has the statistical analysis been performed appropriately and rigorously? 

Reviewer #1: Yes

3. Have the authors made all data underlying the findings in their manuscript fully available?

Reviewer #1: Yes

4. Is the manuscript presented in an intelligible fashion and written in standard English?

Reviewer #1: No

5. Review Comments to the Author

Reviewer #1: This manuscript reports on an experiment in which lay participants sentencing decisions and justifications were compared with professional judges' decisions in response to a child abuse case vignette. They conclude that judges are less motivated by retribution as a sole justification for punishment, and consequently they are more lenient. Overall, the study is interesting and well-justified. I have some comments that might help to polish the paper.

-I don't think your average reader will be familiar with the Kant/Bentham distinction so it would be beneficial to include a sentence or two defining retributive and utilitarian philosophies at the outset.

-There are a few instances in the intro in which the sentences are a bit long and could be broken into multiple sentences for clarity.

-On p. 3 the word "capacitation" appears where I think the authors meant "incapacitation".

-Asking the participants for an open-ended justification seems like a clever strategy to encourage deeper processing of their decision. I do wonder if this promoted utilitarianism, given that it is considered the "cold calculated" decision (see Greene et al.'s 2001 dual process theory).

-I’m assuming the authors used a Bonferroni correction due to family-wise error concerns, but they don’t specify why or how they did this.

-In the discussion, the authors note that because retribution was ranked third in importance that the measure was potentially inaccurate. However, the extant literature suggests that people tend toward utilitarian justifications while motivated by retribution (i.e., they like things like deterrence in theory but the underlying decision-making mechanism is actually retributive; see Carlsmith et al., 2002). For this reason, it's interesting that the authors chose to include a scale that made the justifications mutually exclusive, forcing participants to choose. Lay people tend not to make the same distinctions that philosophers do, even in the face of logical incompatibility.

-Some details about the recruitment method are needed. Which company was used? Does this present any issues with selection bias? Unique demographics?

-Judges are notoriously difficult to recruit, which makes these findings a unique contribution. However, the reader needs more information about the characteristics of the judges - were they from the U.S.? How were they identified?

-How much were participants paid?

6. PLOS authors have the option to publish the peer review history of their article (what does this mean?). If published, this will include your full peer review and any attached files.

Reviewer #1: No

---

## [Author Response · Author response to Decision Letter 0]

21 Sep 2022

Dear Editor,

We apologize for not fully complying with PLOS ONE’s publication criteria. We thank you for indicating the details in this regard. We have thus, now incorporated the additional requirements into the revised version of the manuscript as mentioned below.

1. Please ensure that your manuscript meets PLOS ONE’s style requirements, including those for file naming. The PLOS ONE style templates can be found at 

Response 1: Thank you for your comment. We have ensured that the manuscript complies with the formatting samples that have been provided.

Response 2: Thank you for your comment. We have now added information to the “Methods” section of the manuscript as well as the online submission information. The relevant section in the manuscript can be found at lines 115-118 on page 6. 

Response 3: Thank you for your comment. Written informed consent was obtained electronically as is reflected in lines 127-129 on page 6. 

Dear Reviewer #1,

We are extremely thankful to you for your helpful comments. Your comments provided valuable suggestions for improving the manuscript and furthered our research interests. The language, which received a negative response, has been now enhanced through a professional English editing service.

This manuscript reports on an experiment in which lay participants sentencing decisions and justifications were compared with professional judges’ decisions in response to a child abuse case vignette. They conclude that judges are less motivated by retribution as a sole justification for punishment, and consequently they are more lenient. Overall, the study is interesting and well-justified. I have some comments that might help to polish the paper.

Response: Thank you very much for your comment. We are glad that you appreciated our paper and we do appreciate your valuable comments.

1. I don’t think your average reader will be familiar with the Kant/Bentham distinction so it would be beneficial to include a sentence or two defining retributive and utilitarian philosophies at the outset.

Response 1: Thank you for your comment. As suggested, we have added the following few lines to the beginning of the Introduction:

This study examines “retributivism” versus “utilitarianism”—a jurisprudential issue regarding the justification of punishment—from the perspective of the differences between experts and non-experts. Retributivism implies that the essence of punishment is retribution for crime; and it seeks to offset crime through punishment. Utilitarianism is based on the idea that punishment should serve the function of preventing future crimes. It, therefore, seeks to deter recidivism and potential offenders from committing crimes. 

2. There are a few instances in the intro in which the sentences are a bit long and could be broken into multiple sentences for clarity.

Response 2: Thank you for your comment. We have now reviewed the entire manuscript, including the introduction section, and revised the sentences according to your suggestion. In addition to shortening the sentences, we also revised the manuscript to simplify it by eliminating compound predicates.

3. On p. 3 the word “capacitation” appears where I think the authors meant “incapacitation.”

Response 3: Thank you for pointing it out. Our apologies. This was indeed our mistake. It has now been corrected (line 88 on page 4).

4. Asking the participants for an open-ended justification seems like a clever strategy to encourage deeper processing of their decision. I do wonder if this promoted utilitarianism, given that it is considered the “cold calculated” decision (see Greene et al.’s 2001 dual process theory).

Response 4: Thank you for your comment. As you may have guessed, the open-ended question was aimed to not only draw attention to the vignette but also served as a device to facilitate deeper processing. Your comment gave us an idea of what would happen if this device was not included or if the participants were under a high cognitive load, as in the criminal trial. Therefore, we have supplemented the manuscript with a discussion of this idea (lines 296-301 on page 13).

5. I’m assuming the authors used a Bonferroni correction due to family-wise error concerns, but they don’t specify why or how they did this.

Response 5: Thank you for your comment. The reason and method for the correction have been added to lines 194-196 on page 9: “To correct the family-wise error rate, multiple comparisons were conducted using the Bonferroni correction. The p-values were corrected in order of absolute t-values.”

6. In the discussion, the authors note that because retribution was ranked third in importance that the measure was potentially inaccurate. However, the extant literature suggests that people tend toward utilitarian justifications while motivated by retribution (i.e., they like things like deterrence in theory but the underlying decision-making mechanism is actually retributive; see Carlsmith et al., 2002). For this reason, it’s interesting that the authors chose to include a scale that made the justifications mutually exclusive, forcing participants to choose. Lay people tend not to make the same distinctions that philosophers do, even in the face of logical incompatibility.

Response 6: Thank you for your comment. We completely agree with the point made by you. The struggle between theory and motive is a major concern which motivated this study. We wanted to highlight their attitude of being rational in their emphasis on utilitarian justifications by also using a non-conflicting Likert scale in conjunction. However, the logic is that a conflicting ratio-type scale will force them to choose a genuine (accurate) response. We have thus added one sentence in the discussion section to clarify the same (lines 278-280 on page 12).

7. Some details about the recruitment method are needed. Which company was used? Does this present any issues with selection bias? Unique demographics?

Response 7: Thank you for your comment. We have added details about the recruitment process and a description of the Internet company (lines 113-115 on page 5) used for the survey: “Judges and laypeople were recruited from a panel of 2.2 million people representing the Japanese adult population (20 years and older) registered with the Internet survey company Rakuten Insight, Inc”. We assumed that the effect of selection bias was very weak because the company provided a sample that is representative of the Japanese population by using random sampling method. However, since the sample of judges was small, bias in age and gender was observed.

8. Judges are notoriously difficult to recruit, which makes these findings a unique contribution. However, the reader needs more information about the characteristics of the judges - were they from the U.S.? How were they identified?

Response 8: Thank you for your comment. Please see the first paragraph on page 5 (Materials and Methods section). We used an affiliate of one of the largest retail services in Japan (similar to Amazon). Judges who shop at that service are also registered. We recruited active judges from a very small sample of those registered as “highly qualified professionals.” 

9. How much were participants paid?

Response 9: Thank you for your comment. They were compensated with shopping points worth US$0.2 (lines 124-126 on page 6).

---

## [Decision Letter · Decision Letter 1]

19 Oct 2022

PONE-D-22-13456R1

Comparing sentencing judgments of judges and laypeople: The role of justifications

PLOS ONE

Dear Dr. Watamura,

Thank you for submitting your manuscript to PLOS ONE. After careful consideration, we feel that it has merit but does not fully meet PLOS ONE’s publication criteria as it currently stands. Therefore, we invite you to submit a revised version of the manuscript that addresses the points raised during the review process.

I served as Reviewer 1 in the initial submission and was invited to act as Guest Editor for the second submission. I was pleased with the thorough responses to my suggestions and believe each has been addressed appropriately. Per journal policy, a sole reviewer cannot serve as editor, and so a second reviewer’s feedback was requested. Reviewer 2 provides straightforward, minor recommendations, all of which I believe will enhance the final product and should therefore be addressed in the revision. I have some complementary recommendations which might assist in making these specific changes.

I agree with Reviewer 2’s second point that Kantian ethics are not well known to other professionals. Perhaps it would be helpful, for example on line 43, to simply say something like “Retributivism (which is based on Kant’s moral philosophy)…” then provide a source for a thorough review. Alternatively, you could remove reference to Kant and simply refer to something like “retributive focused philosophies”. Relatedly, for that same passage, I wonder if it’s more advantageous to use a word other than ‘retributive’ in its description, for example: “Retributivism implies that the essence of punishment is to provide just deserts for crime.”

Regarding Reviewer 2’s comment about lines 162-167 on p. 7-8, this passage could indeed benefit from clarification. If I’m not mistaken, it expresses the idea that although punishment justifications are philosophically at odds and often incompatible in practice, Likert-type measures allow participants to rate high agreement with all of them. This strong endorsement of all justifications makes it difficult to identify the relative importance of each. Given your assertion that punishment is more likely a hybrid combination of motivations, in which some are prioritized over others, you forced participants to weight the relative importance via a scale summing 100. Is this interpretation correct? In any case, this passage should be reworded for clarity.

We look forward to receiving your revised manuscript.

Kind regards,

Susan Yamamoto

Guest Editor

PLOS ONE

Journal Requirements:

Reviewers' comments:

Reviewer's Responses to Questions

**Comments to the Author**

1. If the authors have adequately addressed your comments raised in a previous round of review and you feel that this manuscript is now acceptable for publication, you may indicate that here to bypass the “Comments to the Author” section, enter your conflict of interest statement in the “Confidential to Editor” section, and submit your "Accept" recommendation.

Reviewer #2: (No Response)

2. Is the manuscript technically sound, and do the data support the conclusions?

Reviewer #2: Yes

3. Has the statistical analysis been performed appropriately and rigorously? 

Reviewer #2: Yes

4. Have the authors made all data underlying the findings in their manuscript fully available?

Reviewer #2: Yes

5. Is the manuscript presented in an intelligible fashion and written in standard English?

Reviewer #2: Yes

6. Review Comments to the Author

Reviewer #2: In this interesting paper the authors compare layperson (i.e., nonexpert) and judge (i.e., expert) ratings of considerations for justice and sentence length in response to a vignette describing a fatal child abuse case.

Most of my suggestions for the authors are stylistic and there are a few areas where I am unclear on what they are referring to:

1. Some of the claims in the introduction need citations, such as the definitions of utilitarianism and retributivism, and the idea that media might be contributing to people's perceptions of crime, and the fact that people view child abuse as particularly heinous given the combination of lack of parenting duties fulfilled and the vulnerable victims.

2. Some of the concepts could benefit from more explanation. For example, I do not know what a "traditional Kantian moral code" is as someone who is 'adjacent' to your specific research area nor am I sure what "episodic media coverage" means

3. On p. 5, line 95 the authors refer to judges as not being biased. I think it would be more accurate to say they are "trained to be less influenced by bias" because I'm sure some judges are indeed biased.

4. On page 6 when the authors are describing the sample of judges, I am a bit confused - is the entire 50 person sample judges? On line 120-121 they also say "because there were too few judges, we aimed to recruit as many as possible and did not limit the demographic variables". I'm not sure if that means they recruited judges of any demographics or they recruited non-judges. I think just a simple edit of the sentence can make this more clear. Maybe something like "We permitted judges with any type of background demographics to ensure we could recruit 50."

5. p.7-8, likes 162-167 are a bit confusing, I am not entire sure what the authors are saying in these lines

6. I'm wondering why the authors decided to share the prosecutor's recommended sentence of 13 years in the sentencing decision vignette. Is it possible this influenced the responses? What was the reasoning for giving it? I apologize if I missed this detail in the paper.

7. PLOS authors have the option to publish the peer review history of their article (what does this mean?). If published, this will include your full peer review and any attached files.

Reviewer #2: No

---

## [Author Response · Author response to Decision Letter 1]

1 Nov 2022

Thank you for reviewing our manuscript and giving us the opportunity to resubmit it. 

We are very pleased to hear that you are satisfied with our previous response. Your insightful recommendations were in line with our intentions; therefore, we have revised the manuscript to reflect your suggestions.

I served as Reviewer 1 in the initial submission and was invited to act as Guest Editor for the second submission. I was pleased with the thorough responses to my suggestions and believe each has been addressed appropriately. Per journal policy, a sole reviewer cannot serve as editor, and so a second reviewer’s feedback was requested. Reviewer 2 provides straightforward, minor recommendations, all of which I believe will enhance the final product and should therefore be addressed in the revision. I have some complementary recommendations which might assist in making these specific changes.

I agree with Reviewer 2’s second point that Kantian ethics are not well known to other professionals. Perhaps it would be helpful, for example on line 43, to simply say something like “Retributivism (which is based on Kant’s moral philosophy)…” then provide a source for a thorough review. Alternatively, you could remove reference to Kant and simply refer to something like “retributive focused philosophies”. Relatedly, for that same passage, I wonder if it’s more advantageous to use a word other than ‘retributive’ in its description, for example: “Retributivism implies that the essence of punishment is to provide just deserts for crime.”

Response: Thank you for your valuable suggestions. We agree with your assessment that referencing Kant is unnecessary as Kantian ethics may not be well known. Hence, in the revised manuscript, we have deleted mentions of Kant.

Regarding Reviewer 2’s comment about lines 162-167 on p. 7-8, this passage could indeed benefit from clarification. If I’m not mistaken, it expresses the idea that although punishment justifications are philosophically at odds and often incompatible in practice, Likert-type measures allow participants to rate high agreement with all of them. This strong endorsement of all justifications makes it difficult to identify the relative importance of each. Given your assertion that punishment is more likely a hybrid combination of motivations, in which some are prioritized over others, you forced participants to weight the relative importance via a scale summing 100. Is this interpretation correct? In any case, this passage should be reworded for clarity.

Response: Thank you for understanding what we were trying to say. Your interpretation is perfectly correct. We have revised the passage as per your recommendations. (Page 8, Lines 165 to 168)

Comments to the Author

Reviewer #2: In this interesting paper the authors compare layperson (i.e., nonexpert) and judge (i.e., expert) ratings of considerations for justice and sentence length in response to a vignette describing a fatal child abuse case. Most of my suggestions for the authors are stylistic and there are a few areas where I am unclear on what they are referring to:

Response: Thank you for your comments. We apologize for any lack of clarity in sections of the manuscript. We have addressed all your comments as below; these revisions have greatly improved our manuscript.

1. Some of the claims in the introduction need citations, such as the definitions of utilitarianism and retributivism, and the idea that media might be contributing to people's perceptions of crime, and the fact that people view child abuse as particularly heinous given the combination of lack of parenting duties fulfilled and the vulnerable victims.

Response: Thank you for your comment. As per your suggestion, we have inserted citations where needed.

2. Some of the concepts could benefit from more explanation. For example, I do not know what a "traditional Kantian moral code" is as someone who is 'adjacent' to your specific research area nor am I sure what "episodic media coverage" means

Response: Thank you for your comment. We have changed “traditional Kantian moral code” to “retributive focused philosophies” throughout the manuscript in light of the editor's comment. (Page 3, Lines 49) According to Katz et al. (2019), child maltreatment is often taken up as an individual case that is reported on an episodic level rather than on a policy level as a social problem. (Page 4, Lines 83) Additionally, we have added necessary explanations for other concepts.

3. On p. 5, line 95 the authors refer to judges as not being biased. I think it would be more accurate to say they are "trained to be less influenced by bias" because I'm sure some judges are indeed biased.

Response: Thank you for your insightful comment. We have made the revision as you suggested. (Page 5, Line 97)

4. On page 6 when the authors are describing the sample of judges, I am a bit confused - is the entire 50 person sample judges? On line 120-121 they also say "because there were too few judges, we aimed to recruit as many as possible and did not limit the demographic variables". I'm not sure if that means they recruited judges of any demographics or they recruited non-judges. I think just a simple edit of the sentence can make this more clear. Maybe something like "We permitted judges with any type of background demographics to ensure we could recruit 50."

Response: As you surmised, all 50 participants were judges. We have made the correction as you suggested. (Page 6, Lines 122 to 124)

5. p.7-8, likes 162-167 are a bit confusing, I am not entire sure what the authors are saying in these lines

Response: Thank you for your comment. The editor has suggested that this section requires further clarification. The revised section is as follows: (Page 8, Lines 164 to 171).

However, while punishment justifications are philosophically at odds and often incompatible in practice, the Likert scale allows participants to rate high agreement with all of them (i.e., acquiescence-response; [35]). This strong endorsement of all justifications creates difficulties in identifying the relative importance of each [36]. Based on our theoretical definition that the psychological construct of sentencing is a hybrid of multiple justifications [37, 38], we deemed it more appropriate to examine the relative weighting of justifications on a ratio-type scale.

6. I'm wondering why the authors decided to share the prosecutor's recommended sentence of 13 years in the sentencing decision vignette. Is it possible this influenced the responses? What was the reasoning for giving it? I apologize if I missed this detail in the paper.

Response: Thank you for your important comments. In experiments where the severity of punishment is the dependent variable, the lack of a baseline, such as a plea, can lead to extreme responses and hinder the analysis (Watamura et al., 2014). Hence, we shared the plea; 13 years is a plausible plea in this vignette. We have added this explanation to the text to aid clarity. (Page 8, Lines 182 to 184)

---

## [Editor Report · Decision Letter 2]

8 Nov 2022

Comparing sentencing judgments of judges and laypeople: The role of justifications

PONE-D-22-13456R2

Dear Dr. Watamura,

We’re pleased to inform you that your manuscript has been judged scientifically suitable for publication and will be formally accepted for publication once it meets all outstanding technical requirements.

Kind regards,

Susan Yamamoto

Guest Editor

PLOS ONE
---

## [Editor Report · Acceptance letter]

10 Nov 2022

PONE-D-22-13456R2 

Comparing sentencing judgments of judges and laypeople: The role of justifications 

Dear Dr. Watamura:

I'm pleased to inform you that your manuscript has been deemed suitable for publication in PLOS ONE. Congratulations! Your manuscript is now with our production department. 

Kind regards, 

on behalf of

Dr. Susan Yamamoto 

Guest Editor

PLOS ONE